# Analysis of the distraction impact on driving performance across driving styles: A driving simulator study in various speed conditions

Mobina Faqani[1,2¤], Habibollah Nassiri[2*], Mahdi Rezaei[3], Mohsen Ramezani[1]

**1** School of Civil Engineering, the University of Sydney, Sydney, Australia, **2** Civil Engineering Department, Sharif University of Technology, Tehran, Iran, **3** Road Traffic Injury Research Center, Tabriz University of Medical Sciences, Tabriz, Iran

¤ Current Address: Civil Engineering Department, Sharif University of Technology, Tehran, Iran
* nassiri@sharif.edu

## Abstract

Distracted driving is a mounting global issue, prompting numerous naturalistic and simulator-based investigations. This study investigates the impact of hands-free (HF) conversation and texting distractions on driving performance during car-following experiments. Three experiments were designed: a baseline (control) condition, HF conversation, and text messaging. Driving data were collected from 40 participants of driving simulator experiments, conducted under six different speed conditions: (i) free-flow, (ii) coherent moving flow, (iii) synchronized flow, (iv) jam density, (v) recovery from jam density, and (vi) collision avoidance. To analyze driving performance across various mobile phone distracted driving (MPDD) experiments, participants are partitioned into three distinct groups: aggressive, moderate, and conservative, based on their driving styles using k-means clustering. Statistical analyses, including t-tests, Friedman Test, and Wilcoxon Signed-Rank Test, were conducted to evaluate driving performance metrics such as Standard Deviation of Lateral Position (SDLP) across conditions (i)-(iv), Acceleration Reaction Time (ART) in condition (v), and Time to Initial Braking Location (TIBL) in condition (vi).

The findings indicated that HF conversation had no effect on SDLP in the free-flow condition. However, it led to a reduction in SDLP for the conservative group in the coherent moving flow condition, for both moderate and conservative groups in the synchronized flow condition, and for the moderate group in the jam density condition. Additionally, HF conversation was associated with a decrease in ART among conservative participants, while it significantly increased TIBL for both moderate and conservative groups. Conversely, texting led to an increase in SDLP for moderate and conservative participants in the free-flow condition and for the moderate group in the coherent moving flow condition. However, it resulted in a reduction in SDLP for the conservative group in the coherent moving flow condition. Texting had no significant effect on SDLP in the jam density condition or on ART. However, it significantly

**Data availability statement:** The data underlying our study are publicly available in the following GitHub repository: https://github.com/Faqani1993/driving-distraction-performance-styles-data.

**Funding:** The author(s) received no specific funding for this work.

**Competing interests:** The authors have declared that no competing interests exist.

increased TIBL among moderate and conservative participants. These findings can inform legislation, policy development, countermeasures, and future research.

---

## Introduction

Mobile phone distracted driving (MPDD) has emerged as a critical area of concern in road safety, with substantial evidence pointing to its role in increasing the risk of traffic incidents [1–5]. In 2016, distraction contributed to 9% of all fatal crashes, underscoring its detrimental impact on safety [6]. Drivers who read or compose text messages exhibit slower stimulus-response times, lower speeds, poorer lane tracking, shorter headways, and fewer forward glances—factors that together heighten the likelihood of safety-critical events compared with undistracted driving [7–9]. One study reported that phone use lengthened reaction time by roughly 30% and markedly impaired the driver's ability to keep a safe gap to the lead vehicle [7]. Crash-risk estimates range from two- to nine-fold higher for drivers engaged in mobile-phone tasks than for those who are not distracted [10,11]. Data from the Second Strategic Highway Research Program Naturalistic Driving Study (SHRP 2) further show that hand-based phone activities raise crash odds ratios substantially; texting while driving, for instance, increases risk by a factor of about 6.1 [12,13]. The study conducted indicated that there is a relationship between the use of cellular telephones while driving and a subsequent crash with fatalities. The researchers found out that the use of phone for texts and Internet to be positively associated with road crash fatalities with incremental risks of 8.4%, and 54.6%, respectively [14]. Similarly, Perez et al. (2024) concluded that distractions from smartphone apps and social media—combined with reduced situational awareness and risk perception—may significantly elevate crash risk [15].

A sizeable literature also documents how widespread this behaviour is across regions and demographic groups. For instance, research conducted in China found that 84.1% of surveyed drivers (from a sample size of 414) reported engaging in phone conversations at least weekly while driving [16]. In the United States, a comprehensive roadside observation study revealed that 48% of distracted drivers, out of a sample of 3,265, were seen using mobile phones, and 16.6% were observed texting or dialing [17]. In New Zealand, over half of drivers (sample size of 1,057) admitted to sending or reading between 1 to 5 text messages weekly while driving [18]. Among Australian drivers, 61% of a sample of 484 reported engaging in high-risk mobile phone activities, such as texting or browsing, while driving [19]. In the United Kingdom, 29% of a sample of drivers (n = 314) reported making phone calls daily while driving, with 30% reading text messages and 22% sending texts on a daily basis [20]. Similarly, a survey conducted in Tehran, Iran, found that an alarming 88% of respondents engaged in mobile phone use while driving (sample size of 824) [21]. Also, a survey conducted in Mashhad, Iran, revealed that approximately 93% of drivers use their cell phones while driving at least once per week, with 32.5% reporting consistent cell phone use while driving (n = 255) [22].

Addressing the impact of mobile phone distractions on driving safety requires a nuanced examination of driver performance and behavior under various conditions. Extensive reviews in recent years have highlighted the broad scope of research on mobile phone distractions, focusing on how these distractions impact driving performance, with specific effects on behaviors like speed selection, lane deviations, steering control [23], and even gaze behavior [24,25]. Prior research indicates that MPDD can profoundly influence driving dynamics, though the specific effects vary widely across contexts. For instance, while some studies suggest that cognitive distractions increase variability in lateral control, as evidenced by the standard deviation of the lateral position (SDLP) [26–28], others have reported a reduction [29] or negligible impact on SDLP [30,31].

Research on the influence of MPDD under varying driving conditions also shows mixed results; for instance, young drivers have shown significant lane deviation during nighttime cell phone use [32]. Dialing and conversing on mobile phones, whether hands-free (HF) or handheld, have both been associated with elevated mental workload, impaired peripheral detection, and a notable increase in lateral deviation during dialing [33]. The presence of MPDD has also been demonstrated to alter driving patterns across age groups and distraction scenarios. Studies conducted in simulators with drivers of different ages reveal that mobile phone usage during unexpected driving situations increases lateral variability and diverges from typical driving behavior seen in non-distracted conditions [34].

Moreover, findings indicate that engaging in phone conversations while driving significantly elevates cognitive load, which can lead to reduced speed, extended following distances, and increased variability in lane positioning [35]. Similarly, a study leveraging virtual reality (VR) simulations to assess the impact of cellphone positioning on driving performance noted that drivers with cell phones positioned in their lap exhibited more cautious behavior by checking speed and lane position more frequently, with reduced lateral deviation, reduced time spent looking at the cellphone, and increased time in the correct lane [36]. Additional research indicates that drivers using hands-free devices have better vehicle control than those using handheld phones, as measured by steering patterns and lateral positioning [37]. Meanwhile, texting on messaging platforms like WhatsApp has been shown to impair driving performance, particularly among older drivers, with significant increases in SDLP and collision likelihood [38].

There is ongoing debate regarding reaction times when drivers engage in mobile phone conversations. Reaction time represents how quickly a driver can respond to control the vehicle in a particular situation. Certain authors contend that reaction times decrease when drivers engage in mobile phone conversations [26]. Additionally, other studies in this area suggest that under cognitive distraction, reaction times are prolonged compared to situations without distraction [9,27,39–43]. A study investigated braking responses among young drivers while conversing on hands-free and handheld cell phones compared to a baseline condition, revealing significantly slower reaction times, movement times, and total response times in both cell phone conditions as opposed to the baseline, indicating that conversing on cell phones, whether hands-free or handheld, leads to a reduction in the speed of information processing [44]. A study aimed to assess the reaction times of college-age drivers to peripheral traffic events while engaged in mobile phone conversations. Using a driving simulator, it found that distracted driving due to phone conversations led to reaction times over 40% longer compared to un-distracted conditions. Additionally, this impairment was nearly double for provisional license holders [41]. In a series of driving simulator experiments exploring various distractions like mobile phone use, text reading, and watching a DVD while driving, mobile phone conversations significantly increased mental workload and impacted driving behavior, showing reduced speed, increased headway, and lateral position variability [35].

While previous studies have examined either driving style classification or the effects of mobile phone distraction on driving behavior independently, the combined influence of mobile phone distraction on drivers with varying driving styles remains largely underexplored. This study aims to fill this gap by addressing the following research question: How does mobile phone distraction—specifically hands-free (HF) conversations, which primarily involve cognitive distraction, and texting, which primarily involves manual distraction—affect driving performance, including lateral maneuvering across different traffic conditions, Acceleration Reaction Time (ART), and Time to Initial Braking Location (TIBL), both with and without accounting for distinct driving styles (aggressive, moderate, and conservative)?

Aggressive driving typically involves higher speeds, sudden accelerations, abrupt steering, and sharp maneuvers, while conservative driving is characterized by smoother maneuvers and more gradual acceleration and deceleration. A moderate driving style, as the term suggests, sits between these two extremes [45]. Previous studies have categorized drivers based on behavioral observations and driving metrics, such as speed variance, acceleration distribution, and lane deviation [46–48], or through surveys that measure self-reported driving behaviors [49–51]. For this study, driving style classification considers speed, acceleration, and distance to the lead vehicle. Given our research objectives—to examine the effects of mobile phone distraction, to categorize drivers into distinct driving styles, and to compare performance outcomes with and without considering driving style—the study is designed around a car-following task. Participants follow a lead vehicle through six different traffic conditions: free-flow, coherent-moving flow, synchronized flow, traffic jam, recovery from traffic jam, and collision avoidance. We hypothesize that traffic congestion and the relative position of the lead vehicle [52–55] significantly affect driving aggression, making this scenario suitable for exploring the interaction between driving style and distraction type.

The structure of this paper is as follows: Sect Materials and methods presents the experimental setup, data collection, and methods used to analyze MPDD impacts on driving performance, followed by Sect Results, which details the findings and their implications under varying traffic conditions. The final section, Sect Discussion, concludes with key insights, emphasizing the relevance of these findings to road safety and potential future research directions, including enhancements in experimental design and exploration of additional variables influencing driver behavior under distraction.

## Materials and methods

### Participants characteristics

A total of 40 healthy participants neurologically healthy participants with normal vision and hearing abilities, all licensed drivers, were recruited for this study, comprising 7 females and 33 males. The participants' ages ranged from 18 to 42 years (M = 28.68, SD = 6.9), with driving experience ranging from 1 to 22 years (M = 9.1, SD = 6.44). Of the sample, 55% reported using a mobile phone while driving more than three times per week, indicating a relatively high prevalence of mobile phone use in their driving routines. Additionally, 10% of participants reported having been involved in a road accident while using a mobile phone. In a pre-survey, participants rated the potential for mobile phone use to cause distraction and accidents at an average of 3.18 out of 4, indicating a strong awareness of the associated risks.

To minimize potential learning effects among participants, each individual completed three experimental conditions—baseline, hands-free (HF) conversation, and texting—in a randomized order. This randomization ensured that no specific sequence influenced participant performance, maintaining the integrity and validity of the experimental results.

### Apparatus and stimuli

The experiments were conducted using a medium-fidelity driving simulator at the Road Traffic Injury Research Center in Tabriz, Iran. The simulator was equipped with all essential vehicle operation components, including a steering wheel, accelerator, and brake pedals, providing a realistic driving environment for participants (see Fig 1). The simulator features a cabin modeled after the Peugeot 405 with an autogear, providing participants with an immersive and realistic driving experience. To enhance realism, the simulator is equipped with an environmental noise system that replicates authentic road sounds and operational noise.

The driving scenario was projected onto three 43-inch screens strategically placed around the cabin, providing a wide 270 ° horizontal field of view and a vertical field of view 38 ° to 40 ° for the driver. Each screen featured a 1920×1080 pixel resolution, full high-definition capabilities, and high brightness for enhanced clarity. This immersive setup allowed participants to experience a near-surround panoramic vision, closely replicating real-world driving conditions and ensuring a highly realistic simulation environment.

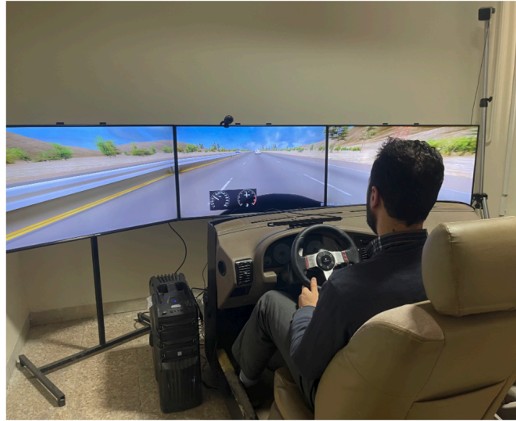
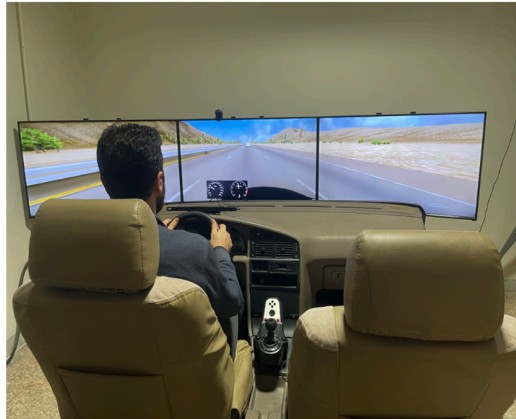

**Fig 1**. **The driving simulator at the Road Traffic Injury Research Center, Tabriz, Iran.**

## Experimental procedure

Participants were eligible for the study if they held a valid driver's license and had at least one year of driving experience. Individuals with a history of neurological disorders, uncorrected vision impairments, or conditions affecting cognitive or motor functions were excluded. Recruitment was conducted through university announcements and social media advertisements.

During the experimental preparation phase, participants were provided with an iPhone 8 and given time to familiarize themselves with the device for texting purposes. They also completed a demographic questionnaire, collecting information such as age, gender, driving experience, and education level. To ensure they were comfortable with the procedure, participants first received verbal instructions and then practiced using the driving simulator for five minutes. This practice session allowed them to become accustomed to the simulator by steering and operating the accelerator and brake pedals. Notably, none of the participants reported experiencing simulator sickness during this phase.

During the experimental trials, participants were instructed to drive according to their natural driving habits, while consistently following the lead vehicle and strictly avoiding any overtaking or lane changes under any circumstances. To minimize external influences, participants were required to refrain from consuming alcohol or caffeinated beverages for 24 hours before the experiment and to ensure adequate rest the night before.

This study was conducted in compliance with the ethical guidelines approved by the Research Ethics Committee of Iran University of Medical Sciences and adhered to the principles outlined in the Declaration of Iran. Written informed consent was obtained from all participants before their involvement in the study. The experimental sessions were conducted between 9:00 a.m. and 12:00 p.m, from 22 February 2023 to 6 May 2023. Participation was voluntary and generally uncompensated; however, participants recruited during their working hours were compensated based on their regular job wages as a token of appreciation.

## Scenario design

The car-following experiment employed a two-factor mixed design: three types of distraction (baseline, hands-free conversation, and texting) and six traffic conditions (free-flow, coherent moving flow, synchronized flow, jam density, recovery from jam density, and collision avoidance). The total driving duration of 574.931 seconds was segmented into six distinct traffic conditions: 19.45 to 64.5 seconds for the first condition, 67.288 to 231.175 seconds for the second, 236.731 to 414.5 seconds for the third, 417.285 to 473.961 seconds for the fourth, 496.733 to 516.19 seconds for the fifth, and

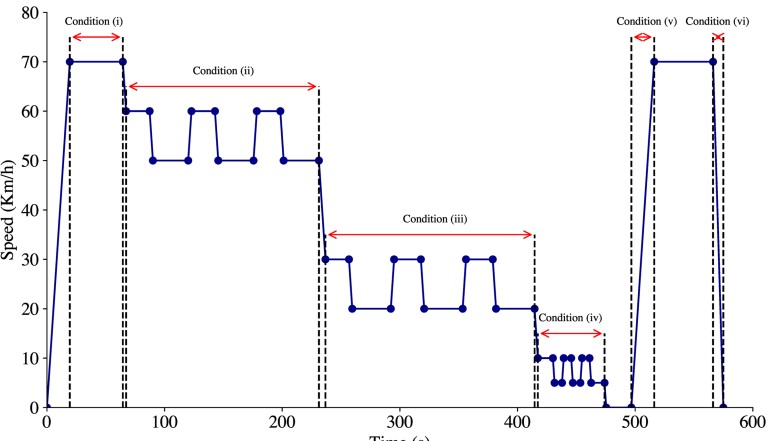

**Fig 2**. Lead vehicle speed profile across different traffic conditions: (i) free-flow; (ii) coherent moving flow; (iii) synchronized flow; (iv) jam density; (v) recovery from jam density; (vi) collision avoidance.

566.171 to 574.931 seconds for the sixth. The speed profile of the lead vehicle under different traffic conditions is illustrated in Fig 2.

The driving scenario was conducted on a 5.1-km two-way straight rural road, featuring three lanes in each direction. Each lane measured 3.75 meters in width, with a 1-meter shoulder on either side. The speed limit was set at 80 km/h, and the weather remained clear and sunny throughout the experiment. To create a controlled yet realistic environment, the road was free of parked cars and pedestrians, with simulated random traffic in the opposite direction and vehicles occasionally passed from the adjacent lanes.

### Cognitive task

The mobile phone tasks were intentionally designed to sufficiently distract drivers and increase their cognitive workload. Two task levels were implemented: hands-free conversation and texting.

In the hands-free conversation task, participants listened to complete arithmetic questions and were required to provide appropriate verbal responses. The arithmetic questions involved calculating the sum of two-digit numbers presented in a random difficulty order. For instance, the sum 67+89 = 156 is considered more challenging, while 10+20 = 30 is regarded as a simpler calculation.

In the texting task, participants received the same arithmetic questions and were required to both compute and text their answers. Similar to the conversation task, the arithmetic problems involved summing two-digit numbers with varying difficulty levels.

Both the hands-free calling and texting tasks were initiated precisely at the beginning of the respective experimental conditions and continued until the end of the scenario, ensuring consistent exposure to cognitive load throughout the task duration.

### Driving data acquisition

Eight key driving data were collected from the simulator at a sampling rate of 50 Hz (see Table 1). These variables capture Speed-related metrics (e.g., speed, Relative speed, Longitudinal acceleration, deceleration), Lane-keeping behavior (e.g., lane deviation, standard deviation of lane position, time spent out of lane), Steering control (e.g., steering angle, steering reversal rate, smoothness of steering), Braking patterns (e.g., brake reaction time, braking force, braking duration), and Headway and following behavior (e.g., time headway, time-to-collision, car-following distance).

**Table 1.** Driving variables recorded from the simulator.

| Variable | Unit |
|---|---|
| Speed | $m/s$ |
| Relative speed | $m/s$ |
| Space headway | $m$ |
| Longitudinal acceleration | $m/s^2$ |
| Degree of acceleration pedal | ° |
| Degree of brake pedal | ° |
| Steering wheel angle | ° |
| Lane deviation | $m$ |

## Methodology

Building on previous research, factors such as congestion [52–54] and the (relative) position of the lead vehicle [55] have been shown to significantly influence driving aggression levels. Therefore, we designed a car-following experiment under six different traffic conditions. Since a aggressive driving style is generally characterized by higher speeds, sudden acceleration and deceleration, abrupt steering angle changes, and harsh lateral and longitudinal maneuvers, whereas a calm driving style is associated with relatively lower speeds, gradual acceleration and deceleration, smooth maneuvers, and mild steering adjustments. A moderate or normal driving style falls between these two extremes [45].

To quantify driving style, we extracted key driving features, including the average, maximum, minimum, 25th percentile, 75th percentile, standard deviation, and variance of eight critical variables (Table 1). For computational efficiency, Min-Max Normalization (Feature Scaling) was applied, transforming the data to a [0,1] range.

Our assumption was that participants exhibited one of two distinct driving styles—aggressive or non-aggressive—under conditions (i) to (iv) in the baseline experiment. The assumption of two clusters was theoretically motivated by the hypothesis that drivers tend to fall into one of two broad behavioral profiles—typically referred to as "conservative" and "aggressive" styles. This choice was further supported by internal validation metrics, including the elbow method, silhouette analysis, and the gap statistic, all of which indicated that $k = 2$ was an appropriate and data-driven choice. Conditions (i)–(iv) were used to distinguish aggressive from non-aggressive drivers because they represent steady-state car-following in free-flow or mildly congested traffic, where headway choice, acceleration, braking and steering are largely voluntary. In contrast, conditions (v) and (vi)—stop-and-go queues and lead-vehicle cut-ins—provoke highly reactive responses that obscure underlying style differences; clustering metrics such as Max-S, Max-ABS-LA and Min-SH did not form distinct groups in these conditions. We therefore excluded conditions (v) and (vi) from style classification and reserved them as performance testbeds for assessing the impact of MPDD, avoiding circular reasoning.

Accordingly, K-means clustering was employed separately for each of these conditions to classify participants into aggressive or non-aggressive driving styles. To ensure methodological robustness, we tested alternative clustering techniques such as hierarchical clustering (with and without PCA) and DBSCAN. However, these methods either resulted in imbalanced groupings, lower silhouette scores, or poor separability. In contrast, K-means with $k = 2$ consistently produced more stable and interpretable clusters. We explored different combinations of driving features across conditions (i) to (iv) to maximize the similarity in driving style classification. Additionally, we assessed cluster quality, stability, and interpretability, with a particular focus on the significance of K-means cluster centers.

Subsequently, we introduced a new participant-specific metric, the degree of conservativeness (Eq 1), defined as:

$$DC = 1 - \frac{1}{4}\sum_{i=1}^{4} DS_i \tag{1}$$

- *DC* = Degree of Conservativeness

- $DS_i$ = Driving Style score in condition i

Based on this metric, participants were categorized as follows: 0 (entirely aggressive), 1 (entirely conservative), and 0.25, 0.5, or 0.75 (moderate drivers). Finally, we evaluated the impact of MPDD on driving performance, specifically analyzing Standard Deviation of Lateral Position (SDLP), Acceleration Reaction Time (ART), and Time to initial Braking Location (TIBL) using the paired t-test and the Wilcoxon Signed-Rank Test (Fig 3).

**Clustering algorithm and statistical overview. K-means Clustering:** K-means clustering is an unsupervised machine learning algorithm used to partition a dataset into *k* clusters based on feature similarity. The algorithm aims to minimize intra-cluster variance by iteratively assigning data points to the nearest cluster centroid and updating the centroids until convergence.

**Paired t-test:** The paired t-test was used to compare driving performance metrics before and after MPDD, assuming the data follows a normal distribution. This test evaluates whether the mean difference between paired observations is statistically significant.

- Null hypothesis ($H_0$): There is no significant difference in the mean values of the driving performance metric between the two conditions (e.g., control vs. distracted).

- Alternative hypothesis ($H_1$): There is a significant difference in the mean values of the driving performance metric between the two conditions.

**Wilcoxon Signed-Rank Test:** Given that normality cannot always be assumed, we also conducted the Wilcoxon Signed-Rank Test as a non-parametric alternative to the paired t-test. It evaluates whether the median difference between paired observations is significantly different from zero.

- Null hypothesis ($H_0$): There is no significant difference between the paired conditions; the median of the differences is zero.

- Alternative hypothesis ($H_1$): There is a significant difference between the paired conditions; the median of the differences is not zero.

**Friedman test:** The Friedman test is a non-parametric statistical test used to detect differences in treatments across multiple related groups. It is an alternative to repeated-measures ANOVA when the assumption of normality is violated.

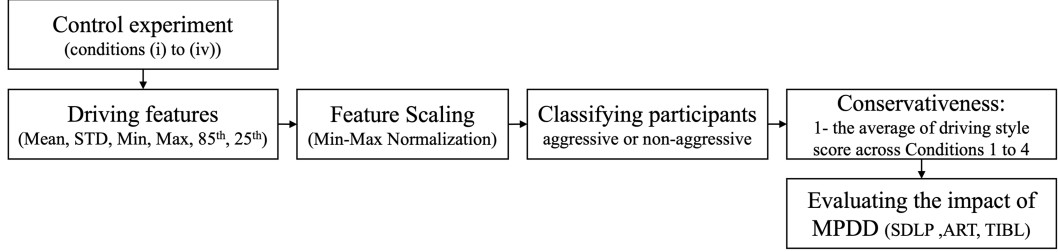

**Fig 3**. **The framework of the evaluation of the MPDD on driving performance based on driving style.**

- Null hypothesis ($H_0$): There is no significant difference between the conditions (the median ranks are the same)).

- Alternative hypothesis ($H_1$): At least one condition differs significantly from the others.

If the *p*-value < a (e.g., 0.05), the null hypothesis is rejected, indicating a significant difference between conditions. If significant, post-hoc pairwise comparisons (e.g., Wilcoxon signed-rank test) are performed to determine which conditions differ.

## Results

### Driving style classification

To classify participants into aggressive and non-aggressive driving groups, we employed the k-means clustering method with k = 2. This classification focused on driving behavior during four traffic conditions (i)-(iv); free-flow, coherent moving flow, synchronized flow, and jam density; observed in the baseline experiment. Key features analyzed in this clustering process included maximum speed (Max-S), maximum absolute value of longitudinal acceleration (Max-ABS-LA), and minimum space-headway (Min-SH), as shown in Fig 4. To ensure consistency in feature dimensions, all variables were normalized prior to clustering.

The results revealed that 15 participants (37.5%) consistently exhibited the same driving style across all four conditions. Meanwhile, 16 participants (40%) maintained the same driving style in three out of the four conditions, and 9 participants (22.5%) displayed an equal mix of both aggressive and non-aggressive driving styles across the four conditions.

Based on our hypothesis that driving styles vary under different traffic conditions, we further evaluated participants' driving styles by averaging their behavior across the four conditions. This analysis indicated that 10% of participants consistently displayed an aggressive driving style, 27.5% exhibited a fully conservative style, and 62.5% demonstrated a moderate driving style. The distribution of driving styles is presented in Fig 5.

### Impact of MPDD on lateral vehicle control

The standard deviation of lateral position (SDLP) is a widely used parameter for assessing the lateral control of vehicles, as highlighted in previous studies [56–59]. To evaluate SDLP in the Mobile Phone Distraction Driving (MPDD) experiments relative to the baseline, we first assessed the normality of the data. If the data followed a normal distribution, a t-test was performed; otherwise, the Wilcoxon Signed-Rank Test was applied. The Friedman test was used to assess whether SDLP significantly differed between the two driving experiments (control and MPDD) within each driving style. When a significant effect was found, post-hoc pairwise comparisons were conducted using the Wilcoxon Signed-Rank Test to determine in which traffic condition (i–iv) the differences occurred. All statistical analyses were performed with a significance level of $p < 0.05$.

Engaging in hands-free (HF) conversation while driving significantly reduced SDLP in coherent, synchronized, and jam-density conditions, suggesting a stabilizing effect on lateral control under these traffic scenarios. Conversely, texting while driving resulted in a significant increase in SDLP during free-flow and coherent moving flow conditions, indicating a marked impact on lateral control (see Fig 6). The results of the Wilcoxon signed-Rank and the t-test are reported in Tables 2, 3, and 4.

Further analysis was performed to evaluate the impact of MPDD on SDLP across different driving styles. The results of the Friedman test are presented in Table 5. Given the presence of a significant effect, post-hoc pairwise comparisons using the Wilcoxon Signed-Rank Test were conducted to identify which specific traffic conditions exhibited significant differences within each driving style. This analysis revealed that engaging in HF conversation significantly reduced SDLP among participants with a moderate driving style under synchronized and jam-density conditions. Similarly, participants with a conservative driving style exhibited a significant decrease in SDLP during coherent moving flow, synchronized

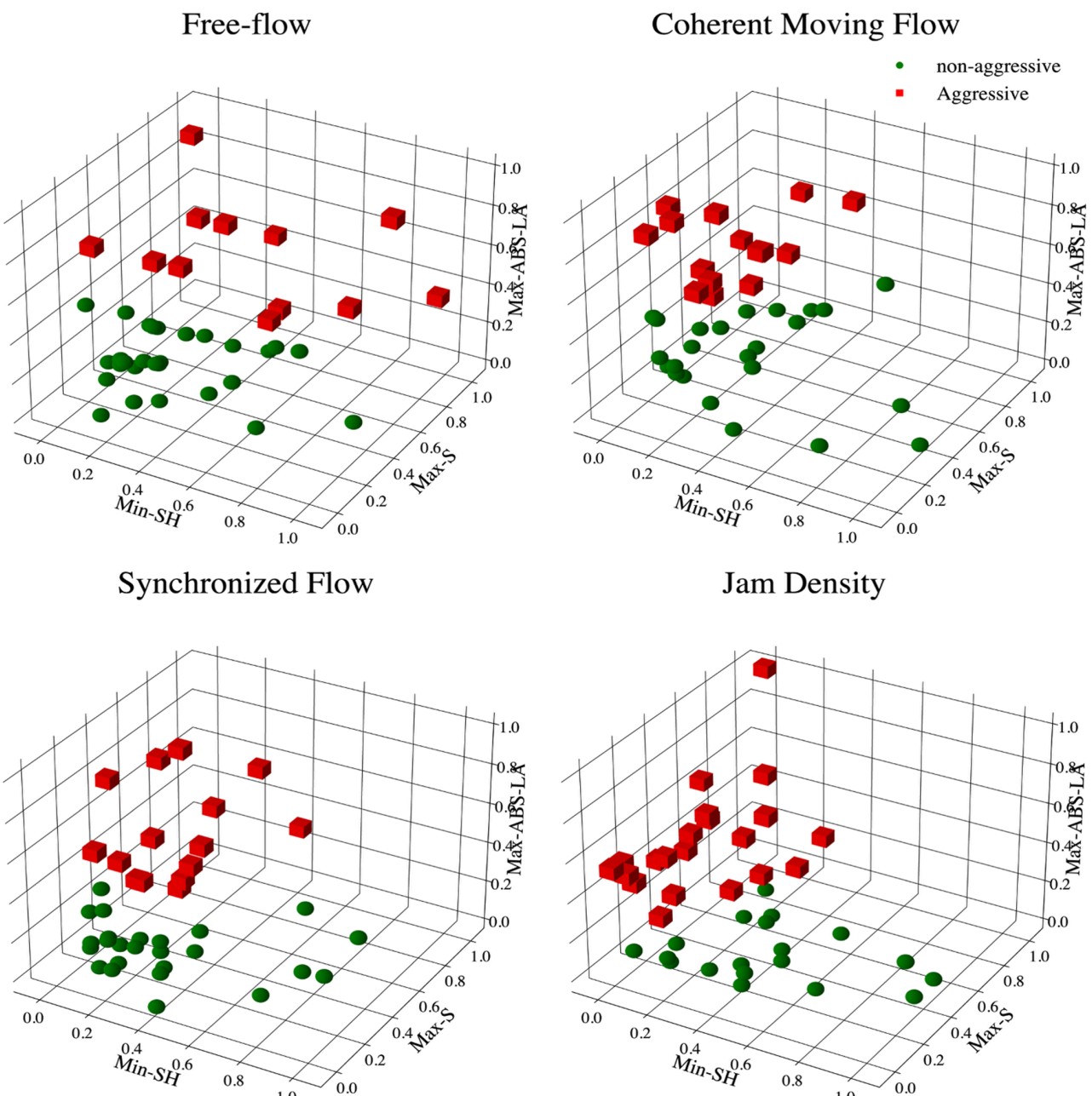

**Fig 4**. **Results of k-means clustering in the baseline experiment (no distraction) across conditions (i)-(iv) (k = 2: aggressive and non-aggressive).**

flow, and jam-density conditions when engaging in HF conversation. In contrast, texting while driving led to a significant increase in SDLP for participants with a moderate driving style in free-flow and coherent moving flow conditions. Among conservative drivers, texting while driving resulted in increased SDLP during free-flow conditions but decreased SDLP during synchronized flow (see Fig 7). The results of the Wilcoxon signed-Rank is reported in Table 6.

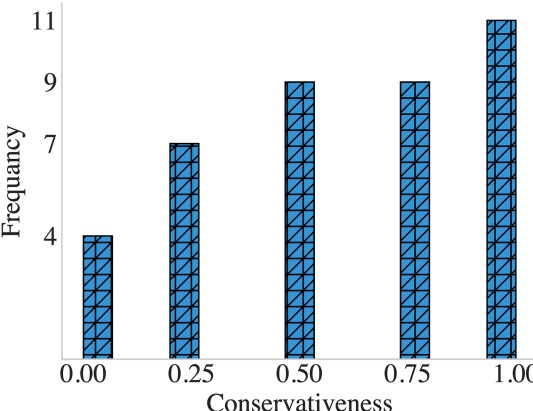

**Fig 5**. Distribution of participants' conservativeness scores (0: aggressive; 0.25, 0.5, and 0.75: moderate; 1: conservative).

The *Mean* SDLP under different cognitive workloads based on driving styles is presented in Fig 8. These results emphasize the significant impact of mobile phone distractions on lateral vehicle control across different traffic conditions and driving styles, with texting posing a particularly pronounced risk to SDLP.

### Impact of MPDD on reaction time

In car-following scenarios, driver reaction time can be characterized by the time taken to respond during acceleration and deceleration maneuvers [60]. In this study, we focus on two specific reaction time metrics. In condition (v), representing recovery from jam density, the primary metric of interest is Acceleration Reaction Time (ART). ART is defined as the duration between a driver's observation of the leading vehicle initiating acceleration and the driver's subsequent activation of the gas pedal to begin accelerating. Condition (vi), which focuses on collision avoidance, is assessed based on Time to Initial Braking Location (TIBL). TIBL is the time it takes for the driver's vehicle to reach the location where the leading vehicle begins braking.

To examine ART and TIBL in the MPDD experiments relative to the baseline, we first assessed the normality of these variables. we first assessed the normality of the data. If the data followed a normal distribution, a t-test was performed; otherwise, the Wilcoxon Signed-Rank Test was applied. The findings indicated that engaging in hands-free (HF) conversation while driving significantly reduced ART, whereas texting did not produce a significant effect. AdditionallyF, MPDD had a significant impact on increasing TIBL, with texting exerting a greater influence. (see Fig 9). The results of the Wilcoxon signed-Rank are reported in Tables 7 and 8.

Further analysis assessed the impact of MPDD on reaction times by driving style, revealing significant differences in ART and TIBL for specific driver groups (see Fig 10). HF conversation notably reduced ART among conservative participants, while MPDD significantly increased TIBL in both moderate and conservative participants. The results of the Wilcoxon signed-Rank Test are reported in Table 9.

The *Mean* ART and TIBL according to driving styles under MPDD conditions are illustrated in Fig 11. In summary, while HF conversation reduces ART, particularly among conservative participants, MPDD significantly increases TIBL, especially in moderate and conservative participants. These findings highlight how texting may affect driver response behaviors differently from HF conversation, particularly in car-following scenarios.

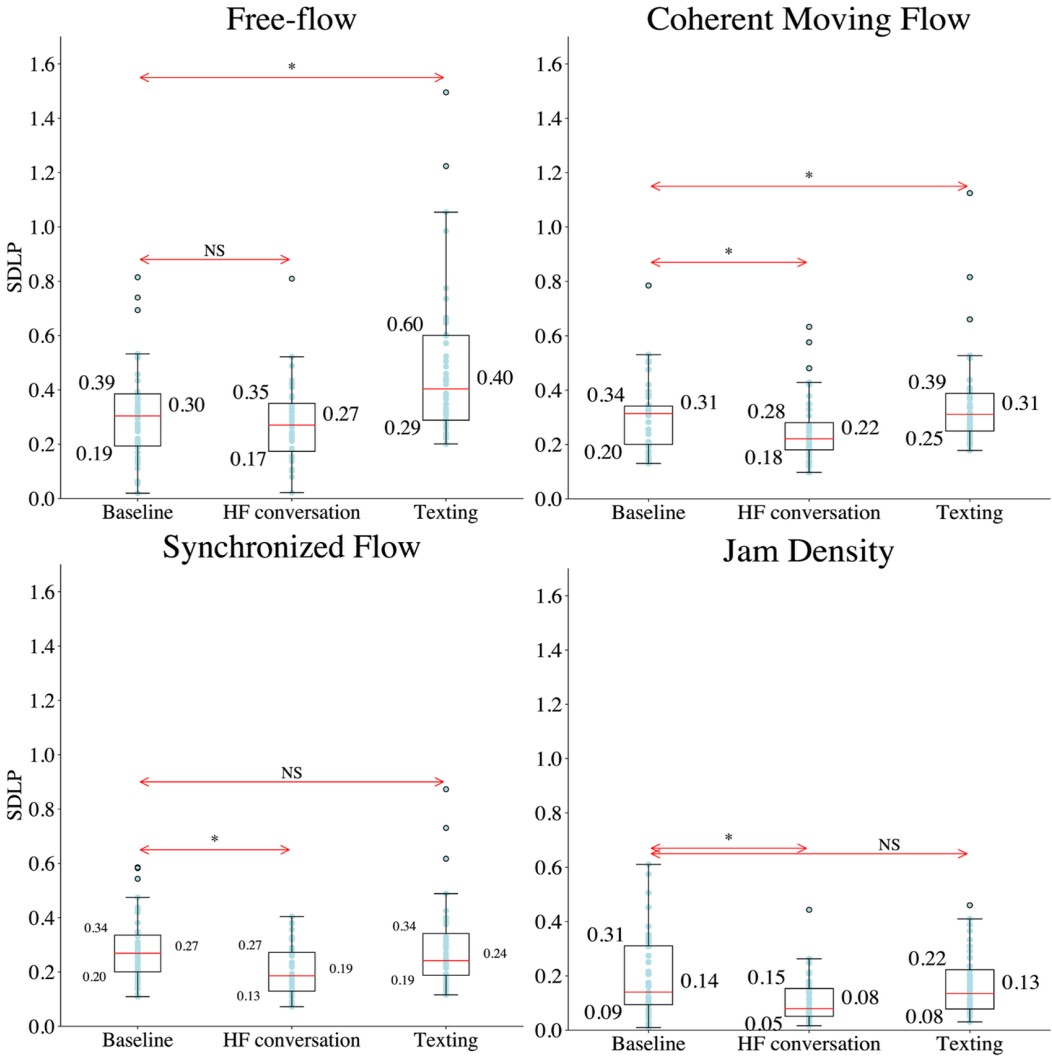

**Fig 6**. **Box plots of SDLP across baseline, HF conversation, and texting for each traffic condition (i–iv).** Horizontal lines indicate statistical comparisons between baseline and MPDD, with significance levels: "NS" (not significant) or "*" (significant at 0.05).

**Table 2**. **Wilcoxon Signed-Rank Test results for the effect of HF conversation on SDLP across different traffic conditions.** Significant differences are indicated by $p < 0.05$.

| Measure | Condition | Control (Median, [IQR]) | HF conversation (Median, [IQR]) | W | z | p-value |
|---------|-----------|-------------------------|----------------------------------|-----|--------|-----------|
| **SDLP** | Free-flow | (0.30, [0.19]) | (0.27, [0.18]) | 344 | −0.887 | 0.38264 |
| | Coherent moving flow | (0.31, [0.14]) | (0.22, [0.10]) | 225 | −2.487 | 0.01202* |
| | jam density | (0.14, [0.22]) | (0.08, [0.10]) | 171 | −3.212 | 0.00094* |

**Table 3**. **T-test results for the effect of HF conversation on SDLP across different traffic conditions.** Significant differences are indicated by $p < 0.05$.

| Measure | Condition | Control (Mean, SD) | HF conversation (Mean, SD) | t(df) | p-value | Effect Size (d) |
|---------|-----------|--------------------|-----------------------------|-------|---------|------------------|
| **SDLP** | Synchronized flow | (0.29, 0.12) | (0.21, 0.09) | 5.50 | 0.00000* | 0.870 |

**Table 4. Wilcoxon Signed-Rank Test results for the effect of texting on SDLP across different traffic conditions.** Significant differences are indicated by *p* < 0.05.

| Measure | Condition | Control (Median, [IQR]) | Texting (Median, [IQR]) | W | z | *p*-value |
|---------|-----------|-------------------------|-------------------------|-----|--------|-----------|
| SDLP | Free-flow | (0.30, [0.19]) | (0.40, [0.31]) | 121 | −3.885 | 0.00004* |
| | Coherent moving flow | (0.31, [0.14]) | (0.31, [0.14]) | 231 | −2.406 | 0.01527* |
| | Synchronized flow | (0.27, [0.14]) | (0.24, [0.15]) | 407 | −0.04 | 0.97349 |
| | Jam density | (0.14, [0.22]) | (0.13, [0.14]) | 350 | −0.806 | 0.42797 |

**Table 5. The Friedman Test results for HF conversation and texting across driving styles.**

| | Driving style | $\chi^2$ (df) | *p*-value | W |
|---------|---------------|-----------|-----------|-------|
| **HF conversation** | Aggressive | 18.75(7) | 0.00901* | 0.67 |
| | Moderate | 59.08(7) | 0.00000* | 0.338 |
| | Conservative | 29.848(7) | 0.0001* | 0.388 |
| **Texting** | Aggressive | 17.991(7) | 0.01201* | 0.643 |
| | Moderate | 78.453(7) | 0.00000* | 0.448 |
| | Conservative | 28.879(7) | 0.00015* | 0.375 |

## Discussion

This paper evaluates the impact of MPDD on drivers' performance, including SDLP, ART, and TIBL. Driving styles were extracted through data normalization, feature selection, clustering, and the definition of degree of conservativeness feature. Additionally, the effect of MPDD on drivers' performance in relation to their driving styles, was assessed. Driving data were collected from 40 participants in driving simulator experiments, encompassing three experimental conditions: a baseline (control) condition, hands-free conversation, and texting. These car-following experiments were conducted under six distinct speed conditions: (i) free-flow, (ii) coherent moving flow, (iii) synchronized flow, (iv) jam density, (v) recovery from jam density, and (vi) collision avoidance. For the evaluation, statistical analyses, including t-tests, the Wilcoxon Signed-Rank Test, and the Friedman test, were performed.

Our findings demonstrate clear, condition- and style-specific effects of MPDD on driving performance. Tables 10 and 11 summarize the influence of distraction types (hands-free vs. texting) on SDLP, ART, and TIBL across aggressive, moderate, and conservative drivers. .

Texting generally led to significant increases in SDLP across several traffic regimes, especially for moderate and conservative drivers under free-flow and coherent-moving flow conditions. Conversely, hands-free conversations had mixed effects: under jam density and synchronized flow, we observed a reduction in SDLP for conservative drivers. These results align with those of Giot et al. (2022) and Rumschlag et al. (2015), who reported that texting degrades lane-keeping more consistently than verbal distractions [56,57].

Interestingly, in specific low-speed, high-density situations, hands-free conversation appeared to stabilize lane control, possibly by increasing cognitive engagement without overloading visual-manual resources. This pattern supports the nuanced findings of Verster and Hartman (2015) and Caird et al. (2018), who suggested that hands-free interactions can sometimes enhance lane centering when drivers self-regulate speed and attention [1,58].

A particularly notable result emerged during the recovery from jam density phase. Here, texting significantly lengthened ART for moderate and conservative drivers, indicating impaired readiness to accelerate following congestion. However, hands-free conversation showed either negligible or decreased ART in this phase for some driver types, especially conservative drivers. This suggests a possible vigilance-enhancing effect of verbal interaction during low-demand traffic conditions.

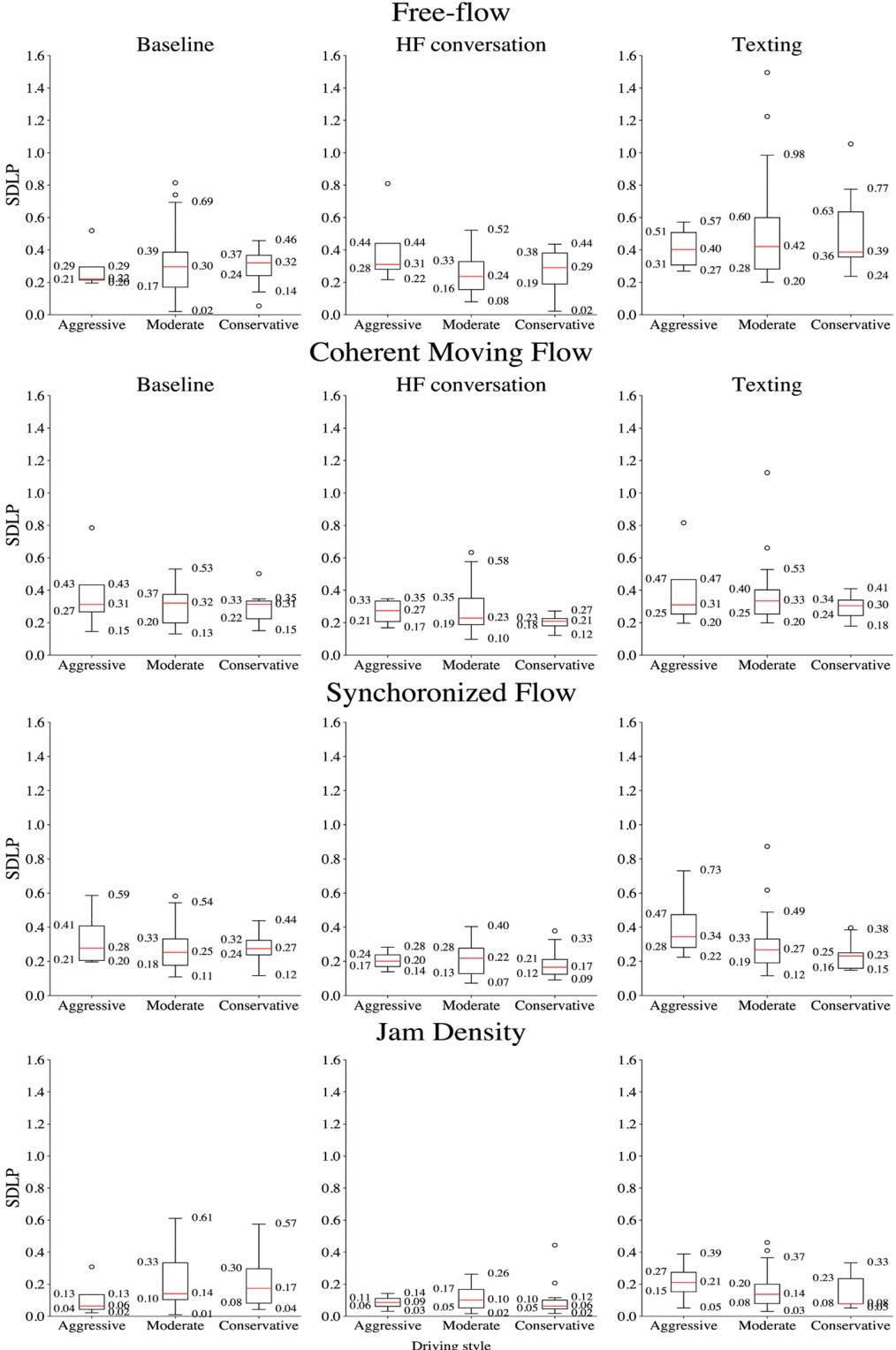

**Fig 7**. **Box plots showing SDLP across experimental conditions, categorized by driving styles.**

**Table 6.** Wilcoxon Signed-Rank Test results for HF conversation and texting across driving styles and traffic conditions.

| | Driving style | Condition | W | z | p-value |
|---|---|---|---|---|---|
| HF conversation | Aggressive | Free-flow | 0.00 | 1.53 | 0.125 |
| | | Coherent moving flow | 3.00 | 0.49 | 0.625 |
| | | Synchronized flow | 0.00 | 1.53 | 0.125 |
| | | Jam density | 5 | 0.00 | 1.000 |
| | Moderate | Free-flow | 113.00 | 1.31 | 0.19081 |
| | | Coherent moving flow | 119.00 | 1.15 | 0.2521 |
| | | Synchronized flow | 56.00 | 2.96 | 0.00309* |
| | | Jam density | 62.00 | 2.77 | 0.00558* |
| | Conservative | Free-flow | 26.00 | 0.558 | 0.57715 |
| | | Coherent moving flow | 4.00 | 2.71 | 0.00684* |
| | | Synchronized flow | 2.00 | 2.98 | 0.00293* |
| | | Jam density | 10.00 | 2.03 | 0.04199* |
| Texting | Aggressive | Free-flow | 0.00 | 1.53 | 0.125 |
| | | Coherent moving flow | 3.00 | 0.49 | 0.625 |
| | | Synchronized flow | 0.00 | 1.53 | 0.125 |
| | | Jam density | 0.00 | 1.53 | 0.125 |
| | Moderate | Free-flow | 63.00 | 2.74 | 0.00613* |
| | | Coherent moving flow | 73.00 | 2.44 | 0.01472* |
| | | Synchronized flow | 132.00 | 0.80 | 0.42614 |
| | | Jam density | 121.00 | 1.10 | 0.27518 |
| | Conservative | Free-flow | 5.00 | 2.58 | 0.00977* |
| | | Coherent moving flow | 28.00 | 0.39 | 0.7002 |
| | | Synchronized flow | 6.00 | 2.47 | 0.01367* |
| | | Jam density | 23.00 | 0.82 | 0.41309 |

These outcomes partially diverge from the aggregate findings of Caird et al. (2018), whose meta-analysis showed consistent reaction time impairments for all phone use types [1]. Our traffic-state-specific approach may account for this discrepancy by isolating traffic regimes where task demands interact differently with cognitive distraction.

TIBL during the collision avoidance phase was significantly impacted by texting across most driver categories, particularly moderate and conservative participants. While hands-free conversation had little effect on TIBL, the significant increases seen during texting align with prior studies showing degraded braking and hazard anticipation during visual-manual phone use [61–63].

Our findings corroborate previous studies that identified texting as a major threat to lane-keeping and reaction time performance [32,64]. For instance, Yan et al. (2022) found a pronounced SDLP increase on straight segments during texting. Similarly, Chouhan et al. (2025) quantified texting as raising crash probability by over 27% due to reaction-time impairments. Moreover, studies like Ortega et al. (2021) and Andrikopoulou et al. (2025) demonstrated that mobile phone use worsens lateral deviation and reduces driver control, particularly among younger drivers or during multitasking events [63,65].

Driving style emerged as a critical moderating factor. moderate drivers exhibited the highest texting-related increases in SDLP and ART, likely due to inherently higher speed variability and less cautious behavior. On the other hand, conservative drivers showed more pronounced sensitivity to hands-free conversations, particularly in lane-keeping heightened self-regulation under dual-task demands.

This style-specific pattern is consistent with Zhang et al.(2024), who found that conservative drivers displayed better takeover control in autonomous vehicles under distraction, while aggressive drivers had longer reaction times and lower braking intensity in urgent situations [66,67].

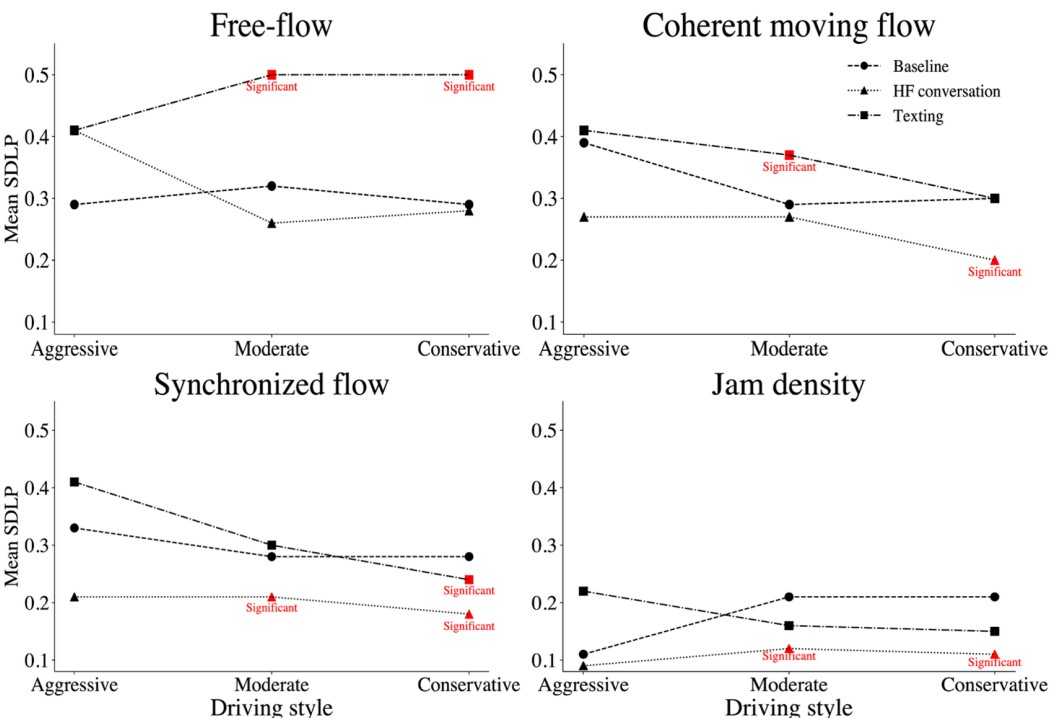

**Fig 8**. **Mean SDLP under MPDD, stratified by driving styles across traffic conditions (i–iv).** Red points denote statistically significant differences compared to the control condition.

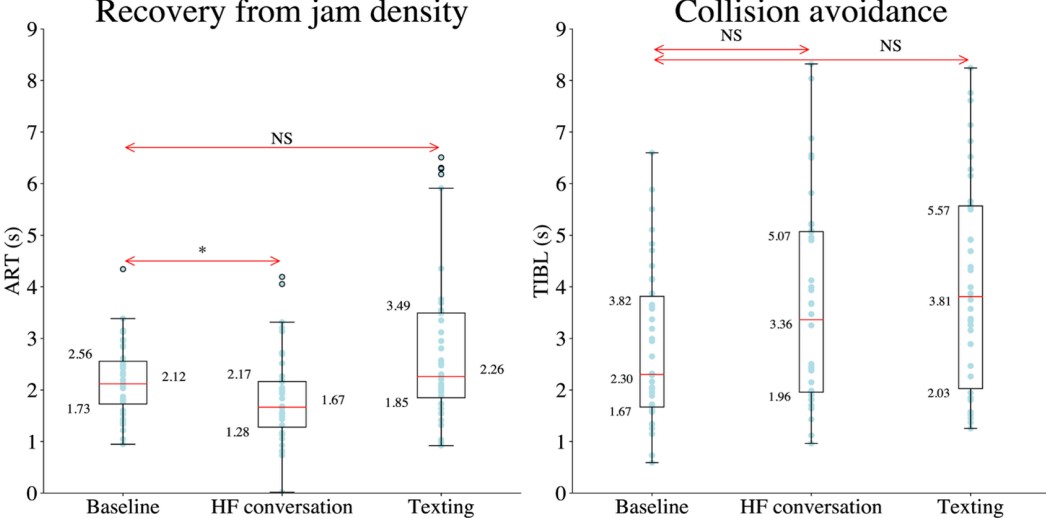

**Fig 9**. **Box plots with individual data points visualizing the distribution of reaction times across MPDD experiments (NS means not significant at 0.05 and \* means significant at 0.05).**

**Table 7**. Wilcoxon Signed-Rank Test results for the effect of HF conversation on ART and TIBL (*$p < 0.05$).

| Measure | Control (Median, [IQR]) | HF Conversation (Median, [IQR]) | W | z | p-value |
|---------|------------------------|--------------------------------|-----|-------|---------|
| ART | (2.12, [0.83]) | (1.67, [0.88]) | 211 | −2.68 | 0.00666* |
| TIBL | (2.16, [1.97]) | (3.25, [3.08]) | 123 | −3.15 | 0.00117* |

**Table 8**. Wilcoxon Signed-Rank Test results for the effect of texting on ART and TIBL (*$p < 0.05$).

| Measure | Control (Median, [IQR]) | Texting (Median, [IQR]) | W | z | p-value |
|---------|------------------------|-------------------------|-----|-------|---------|
| ART | (2.12, [0.83]) | (2.26, [1.64]) | 299 | −1.49 | 0.139 |
| TIBL | (2.16, [1.97]) | (3.57, [3.32]) | 97 | −3.57 | 0.00018* |

The differentiated effects of MPDD on lateral and longitudinal control based on driving style highlight the potential for personalized safety interventions. Advanced driver-monitoring systems (DMS) could improve risk prediction by incorporating behavioral style metrics (e.g., headway, speed variability) alongside distraction detection.

## Limitation and future work

The current research has certain limitations that suggest potential directions for future studies. The study includes a total of 40 participants, which, based on previous research, is a reasonable sample size [1]. However, only four participants exhibited aggressive driving behavior, which may be insufficient for robust statistical analysis. As a result, detecting significant effects within this subgroup was challenging. Furthermore, the experiment is limited to a specific scenario, focusing solely on the impact of mobile phone use and a cognitive task on driving performance. Future research should consider a larger and more diverse sample, incorporating additional behavioral patterns and broader driving scenarios to enhance the generalizability of the findings. Task order was counterbalanced through random assignment to mitigate learning effects; however, future studies should employ a Latin-square experimental design, which offers a more rigorous and standardized means of controlling sequence effects. While this study utilized a medium-fidelity driving simulator, the use of high-fidelity simulators in future work may enhance validity. Although driving style in the present study was identified using performance-based metrics such as SDLP, ART, and TIBL, we acknowledge that gaze behaviors also represent an important dimension of driving style. Integrating an eye-tracking system into future experiments will enable a more detailed analysis of drivers' visual scanning patterns, including gaze shifts, blink rates, and fixation durations, which are critical for assessing distraction and driving safety. Moreover, future studies could integrate gaze-based measures alongside performance indicators to provide a more comprehensive characterization of driving style [68]. Examining the relationship between these visual behaviors and driving performance under various mobile phone usage conditions will provide deeper insights into the mechanisms of driver distraction, ultimately contributing to the development of more effective safety interventions.

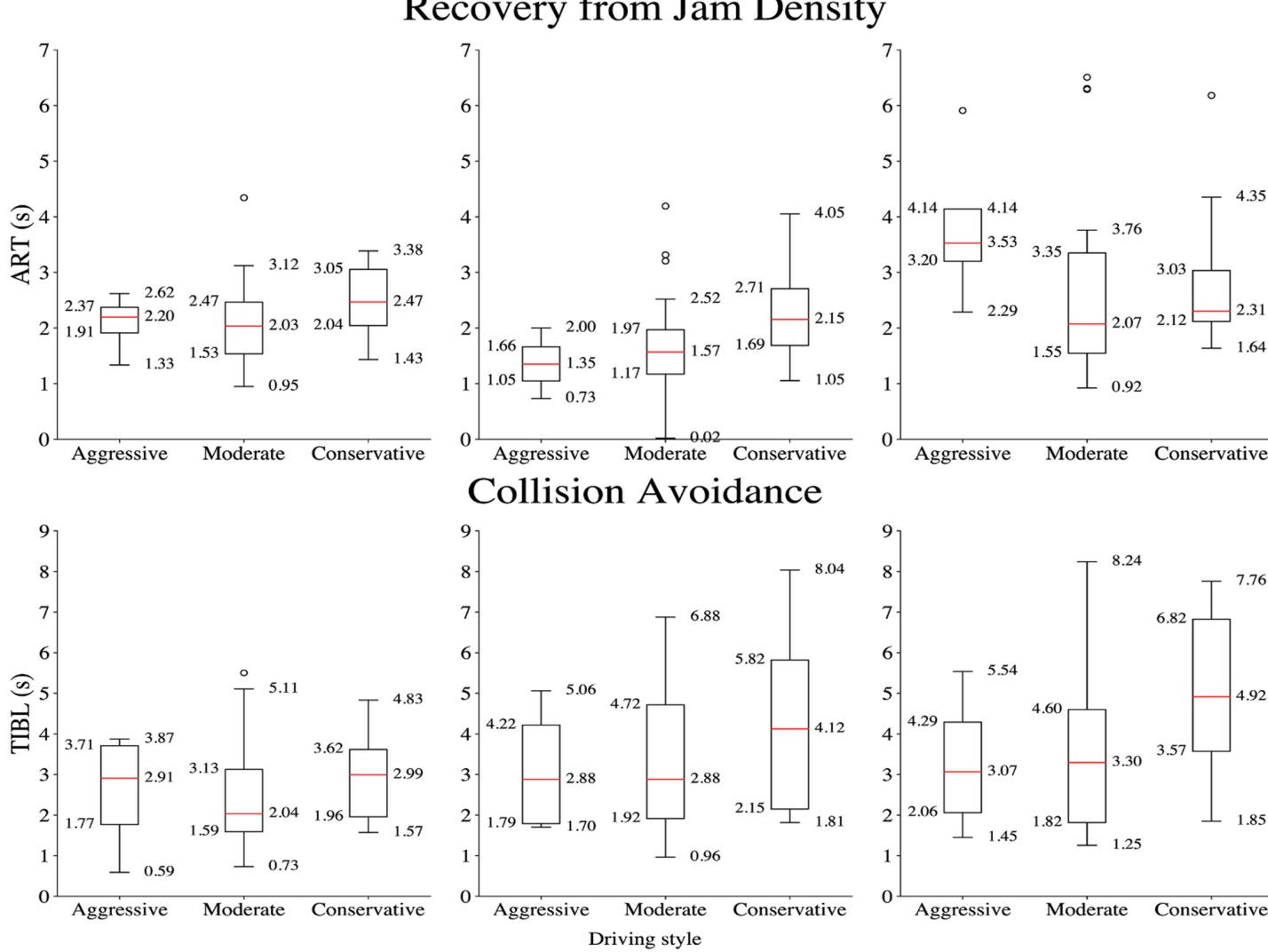

**Fig 10**. **Box plots showing the ART and TIBLs across experiments, categorized by driving styles.**

**Table 9**. Wilcoxon Signed-Rank Test results for HF conversation and texting.

|  |  | Driving style | W | z | *p*-value |
|---|---|---|---|---|---|
| ART | HF conversation | Aggressive | 1.00 | −1.46 | 0.250 |
|  |  | Moderate | 101.00 | −1.66 | 0.1014 |
|  |  | Conservative | 11.00 | −1.96 | 0.05371* |
|  | Texting | Aggressive | 0.00 | −1.83 | 0.125 |
|  |  | Moderate | 137.00 | −0.69 | 0.5077 |
|  |  | Conservative | 32.00 | −0.09 | 0.96582 |
| TIBL | HF conversation | Aggressive | 2.00 | −1.10 | 0.375 |
|  |  | Moderate | 60.00 | −2.16 | 0.03012* |
|  |  | Conservative | 6.00 | −1.96 | 0.05469* |
|  | Texting | Aggressive | 2.00 | −1.10 | 0.375 |
|  |  | Moderate | 50.00 | −2.49 | 0.01147* |
|  |  | Conservative | 2.00 | −2.43 | 0.01172* |

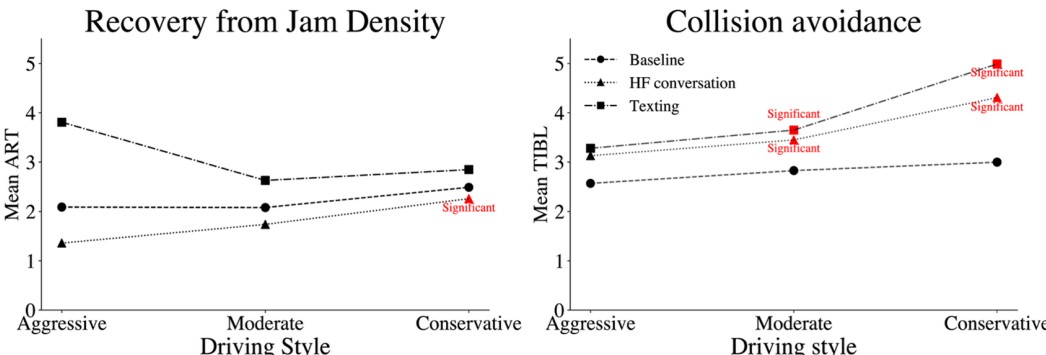

**Fig 11**. ***Mean* reaction time by driving style under MPDD.** Red points denote statistically significant differences in MPDD compared to the control condition.

**Table 10**. Summary of the effect of MPDD on SDLP across all participants, and also participants categorized by driving styles ("NS" (no significant), "+" (significant increase), and "-" (significant decrease)).

|  |  | HF conversation | | | Texting | | |
|---|---|---|---|---|---|---|---|
|  |  | Aggressive | Moderate | Conservative | Aggressive | Moderate | Conservative |
| SDLP | Free Flow | NS | | | + | | |
|  |  | NS | NS | NS | NS | + | + |
|  | Coherent moving flow | − | | | + | | |
|  |  | NS | NS | − | NS | + | NS |
|  | Synchronized flow | − | | | NS | | |
|  |  | NS | − | − | NS | NS | − |
|  | Jam density | − | | | NS | | |
|  |  | NS | − | − | NS | NS | NS |

## Acknowledgment

We gratefully acknowledge the support of the Sharif University of Technology, the Traffic Injury Research Center, and the University of Sydney in making this research possible.

**Table 11. Summary of the effect of MPDD on ART and TIBL across all participants, and also participants categorized by driving styles ("NS" (no significant), "+" (significant increase), and "-" (significant decrease).**

| | | HF conversation | | | Texting | | |
|---|---|---|---|---|---|---|---|
| | | Aggressive | Moderate | Conservative | Aggressive | Moderate | Conservative |
| ART | Recovery from jam density | – | | | + | | |
| | | NS | NS | – | NS | + | + |
| TIBL | Collision avoidance | NS | | | + | | |
| | | NS | NS | NS | NS | + | + |

## Author contributions

**Conceptualization:** Mobina Faqani, Mohsen Ramezani.

**Data curation:** Mobina Faqani, Mahdi Rezaei.

**Formal analysis:** Mobina Faqani.

**Funding acquisition:** Mobina Faqani, Mahdi Rezaei.

**Investigation:** Mobina Faqani.

**Methodology:** Mobina Faqani, Mohsen Ramezani.

**Project administration:** Habibollah Nassiri.

**Software:** Mahdi Rezaei, Mohsen Ramezani.

**Supervision:** Habibollah Nassiri, Mohsen Ramezani.

**Validation:** Mobina Faqani, Mohsen Ramezani.

**Visualization:** Mobina Faqani, Mohsen Ramezani.

**Writing – original draft:** Mobina Faqani.

**Writing – review & editing:** Habibollah Nassiri, Mohsen Ramezani.

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
