## [Decision Letter · Decision Letter 0]

18 Jun 2025

PONE-D-25-19007Analysis of the distraction impact on driving performance across driving styles: A driving simulator study in various speed conditionsPLOS ONE

Dear Dr. Nassiri,

Thank you for submitting your manuscript to PLOS ONE. After careful consideration, we feel that it has merit but does not fully meet PLOS ONE’s publication criteria as it currently stands. Therefore, we invite you to submit a revised version of the manuscript that addresses the points raised during the review process.

Please try to revise your manuscript and respond to all the reviewers' comments.

We look forward to receiving your revised manuscript.

Kind regards,

Quan Yuan, Ph.D.

Academic Editor

PLOS ONE

Journal Requirements: 

2. In the online submission form, you indicated that [The data cannot be shared publicly because they contain confidential information from participants. However, data are available from the corresponding author upon reasonable request for researchers who meet the criteria for access to confidential data.].

[This research was funded by the Sharif University of Technology, Tehran, Iran; the Traffic Injury Research Center, Tabriz, Iran; and the University of Sydney, Sydney, Australia. The authors declare that there are no conflicts of interest related to the research, authorship, or publication of this paper. The data underlying this study are available from the corresponding author upon reasonable request, subject to confidentiality agreements and ethical considerations.]

 [The author(s) received no specific funding for this work.]

4. We are unable to open your Figure file [fig.eps, sample.bib and plos2015.bst]. Please kindly revise as necessary and re-upload.

Reviewers' comments:

Reviewer's Responses to Questions

**Comments to the Author**

1. Is the manuscript technically sound, and do the data support the conclusions?

Reviewer #1: Yes

Reviewer #2: Partly

2. Has the statistical analysis been performed appropriately and rigorously?

Reviewer #1: Yes

Reviewer #2: No

3. Have the authors made all data underlying the findings in their manuscript fully available?

Reviewer #1: Yes

Reviewer #2: No

4. Is the manuscript presented in an intelligible fashion and written in standard English?

Reviewer #1: Yes

Reviewer #2: Yes

5. Review Comments to the Author

Reviewer #1: This paper focuses on a interesting and meaningful topic, where the intersection between driving style and driving performance under MPDD conditions were investigated. However, there are several issues to be clarified.

1. Page 4, Lines 96-100. It would be better to describe more information of participants, such as the accident experience, vision and hearing conditions.

2. Page 4, Line 101. The authors adopted randomized order to minimize potential learning effects. Although the randomized order is valid to some extent, the use of a Latin-square experimental design is more standardized and effective.

3. Page 4, Line 106. The driving simulator shown in Fig 1 does not appear to have motion simulation capabilities and is therefore not a high-fidelity driving simulator. In the field of driving behavior analysis, driving simulators that only have visual and auditory simulations are usually referred to as medium-fidelity driving simulators.

4. Page 5, Line 132-133. The instructions for the participants need to be further clarified. For example, the participants were required to follow the lead vehicle and were not allowed to overtake the vehicle or change lanes under any circumstances.

5. Page 6, Table 1. The initial case form of the variables in the table is not standardized.

6. Page 7, Lines 187-193. Why the authors only adopt conditions (i) to (iv) to recognize participants driving style? There were six condition in total.

Reviewer #2: Although the topic of this manuscript holds certain practical significance and the authors have made efforts in reviewing the relevant literature, there remain notable shortcomings. Specifically, the research objectives and innovative contributions are not clearly articulated; the experimental design suffers from potential confounding factors and lacks sufficient detail; and the definition and interpretation of key variables are not rigorous.

1. In the Introduction, the authors state that mobile phone-related distracted driving (MPDD) has become a key concern in road traffic safety. However, the cited references [6]–[12] mainly document the co-occurrence of distracted driving and MPDD across different countries, without providing concrete evidence on how MPDD specifically increases crash risk. It is recommended that more targeted literature be included to clarify the specific mechanisms through which MPDD impacts driving safety.

2. Despite conducting a relatively comprehensive review of existing studies, the Introduction fails to extract the core research question and clearly define the study's objectives. This lack of focus obscures the novelty and significance of the research. The authors are advised to better articulate the research motivation and explicitly state the scientific problem this study aims to address, along with its potential contributions.

3. The experiment involves six types of traffic conditions, yet each condition lasts only for a short period, and the vehicle’s state changes continuously. This setup may introduce interference effects that compromise the drivers’ ability to perform the tasks as intended, thereby undermining the reliability of the collected data. The experimental design should be refined, or the strategies used to mitigate such interference should be clarified. Additionally, Figure 2 lacks necessary details; it is recommended to include the time–speed profiles at key event points to improve clarity.

4. The current description of the mobile phone task is insufficient to evaluate whether it effectively induces cognitive distraction. Detailed information should be provided on the experimental procedures, such as how distraction tasks are triggered, the content and duration of phone calls, the type and demands of text messaging tasks, etc. Moreover, the method for quantifying cognitive load should be specified, and evidence should be presented to confirm that the distraction task significantly increases cognitive demand on the driver.

5. The study classifies participants into two driving style groups. However, the criteria for this classification are unclear, and the rationale for choosing the K-means clustering method has not been justified. The authors should explain why K-means is suitable for this context and clarify its methodological advantages in this study.

6. The Discussion section is overly brief and mainly repeats the results without engaging in comparative analysis with existing literature. It is recommended to expand this section by contrasting the study’s findings with prior research and analyzing possible reasons for any discrepancies, in order to highlight the scientific value and implications of the work.

6. PLOS authors have the option to publish the peer review history of their article (what does this mean?). If published, this will include your full peer review and any attached files.

Reviewer #1: No

Reviewer #2: No

---

## [Author Response · Author response to Decision Letter 1]

6 Aug 2025

Dear Editorial Office,

We have thoroughly addressed all reviewer comments in the uploaded rebuttal letter and responded to the editor's remarks within the cover letter.

Kindly refer to the documents titled "rebuttal_letter" and "new_cover_letter" for our detailed, point-by-point responses.

Sincerely,

Habibollah Nassiri

Associate Professor of Civil Engineering

Sharif University of Technology

nassiri@sharif.edu

---

## [Decision Letter · Decision Letter 1]

14 Sep 2025

PONE-D-25-19007R1Analysis of the distraction impact on driving performance across driving styles: A driving simulator study in various speed conditionsPLOS ONE

Dear Dr. Nassiri,

Thank you for submitting your manuscript to PLOS ONE. After careful consideration, we feel that it has merit but does not fully meet PLOS ONE’s publication criteria as it currently stands. Therefore, we invite you to submit a revised version of the manuscript that addresses the points raised during the review process.

Please address the reviewer's concern and revise the manuscript again.

We look forward to receiving your revised manuscript.

Kind regards,

Quan Yuan, Ph.D.

Academic Editor

PLOS ONE

Journal Requirements:

Reviewers' comments:

Reviewer's Responses to Questions

**Comments to the Author**

1. If the authors have adequately addressed your comments raised in a previous round of review and you feel that this manuscript is now acceptable for publication, you may indicate that here to bypass the “Comments to the Author” section, enter your conflict of interest statement in the “Confidential to Editor” section, and submit your "Accept" recommendation.

Reviewer #1: All comments have been addressed

Reviewer #2: (No Response)

2. Is the manuscript technically sound, and do the data support the conclusions?

Reviewer #1: Yes

Reviewer #2: Yes

3. Has the statistical analysis been performed appropriately and rigorously?

Reviewer #1: Yes

Reviewer #2: Yes

4. Have the authors made all data underlying the findings in their manuscript fully available?

Reviewer #1: Yes

Reviewer #2: Yes

5. Is the manuscript presented in an intelligible fashion and written in standard English?

Reviewer #1: Yes

Reviewer #2: Yes

6. Review Comments to the Author

Reviewer #1: Thank you for your rely and revisions. My previous concerns have been solved.

However, after reading the revised manuscript, I found that there is a new limiation. Although the authors adopted several metrics (e.g., SDLP, ART, and TIBL) to identify driving style, drivers' gaze behaviors are also important aspects of driving style. Thus, I recommend the authors to clarify that. You can refer to this article: Lane changing maneuver prediction by using driver’s spatio-temporal gaze attention inputs for naturalistic driving

Reviewer #2: (No Response)

7. PLOS authors have the option to publish the peer review history of their article (what does this mean?). If published, this will include your full peer review and any attached files.

Reviewer #1: No

Reviewer #2: No

---

## [Author Response · Author response to Decision Letter 2]

25 Sep 2025

We would like to sincerely thank you for your time and thoughtful consideration of our manuscript. We have carefully addressed all of your comments and suggestions in the 'Response to Reviewers' file.

---

## [Decision Letter · Decision Letter 2]

28 Oct 2025

Analysis of the distraction impact on driving performance across driving styles: A driving simulator study in various speed conditions

PONE-D-25-19007R2

Dear Dr. Nassiri,

We’re pleased to inform you that your manuscript has been judged scientifically suitable for publication and will be formally accepted for publication once it meets all outstanding technical requirements.

Kind regards,

Quan Yuan, Ph.D.

Academic Editor

PLOS ONE

Additional Editor Comments (optional):

Reviewers' comments:

Reviewer's Responses to Questions

**Comments to the Author**

1. If the authors have adequately addressed your comments raised in a previous round of review and you feel that this manuscript is now acceptable for publication, you may indicate that here to bypass the “Comments to the Author” section, enter your conflict of interest statement in the “Confidential to Editor” section, and submit your "Accept" recommendation.

Reviewer #1: All comments have been addressed

2. Is the manuscript technically sound, and do the data support the conclusions?

Reviewer #1: Yes

3. Has the statistical analysis been performed appropriately and rigorously?

Reviewer #1: Yes

4. Have the authors made all data underlying the findings in their manuscript fully available?

Reviewer #1: Yes

5. Is the manuscript presented in an intelligible fashion and written in standard English?

Reviewer #1: Yes

6. Review Comments to the Author

Reviewer #1: Thank you for your rely and revisions. My previous concerns have been solved. The manuscript is suitable to publish now.

7. PLOS authors have the option to publish the peer review history of their article (what does this mean?). If published, this will include your full peer review and any attached files.

Reviewer #1: No

---

## [Editor Report · Acceptance letter]

PONE-D-25-19007R2

PLOS ONE

Dear Dr. Nassiri,

I'm pleased to inform you that your manuscript has been deemed suitable for publication in PLOS ONE. Congratulations! Your manuscript is now being handed over to our production team.

Kind regards,

on behalf of

Dr. Quan Yuan

Academic Editor

PLOS ONE